# Neuropathic Pain Component in Patients with Cervical Radicular Pain: A Single-Center Retrospective Study

**DOI:** 10.3390/medicina58091191

**Published:** 2022-09-01

**Authors:** Jiyeon Kwon, Daeseok Oh, Byeongcheol Lee, Hyunseong Lee, Myoungjin Ko, Sungho Moon, Yeiheum Park, Sehun Kim, Sunyoung Kim

**Affiliations:** Department of Anesthesia and Pain Medicine, Inje University Haeundae Paik Hospital, 875, Haeundae-ro, Haeundae-gu, Busan 612-896, Korea

**Keywords:** cervical radiculopathy, epidural injection, neuropathic pain, spinal stenosis

## Abstract

*Background and Objectives*: Evidence regarding the prevalence of neuropathic pain in patients with cervical radicular pain is limited. This study aimed to investigate the prevalence of neuropathic pain components in patients with cervical radicular pain using established screening tools and identify the relationship between neuropathic pain components and clinical factors. *Materials and Methods*: Data from 103 patients (aged ≥ 20 years) with cervical radicular pain who visited our pain clinic were analyzed retrospectively. Demographic characteristics, history of neck surgery, pain intensity using numeric rating score, dominant pain site, duration of symptoms, and neck disability index were assessed. The prevalence of neuropathic pain components was defined according to the Douleur Neuropathique 4 questions and painDETECT questionnaire tools. Patient characteristics were compared using the chi-square test or Fisher’s exact test for categorical variables and the independent *t*-test or Mann–Whitney U test for continuous variables. The correlation between neck disability index and other variables was analyzed using Pearson’s correlation coefficient. *Results*: Of the 103 patients, 29 (28.1%) had neuropathic pain components. The neck disability index was significantly higher (*p* < 0.001) for patients in the neuropathic pain group (23.79 ± 6.35) than that in the non- neuropathic pain group (18.43 ± 7.68). The Douleur Neuropathique 4 questions (r = 0.221, *p* < 0.025) and painDETECT questionnaire (r = 0.368, *p* < 0.001) scores positively correlated with the neck disability index score. *Conclusions*: The prevalence of neuropathic pain components in patients with cervical radicular pain was low. The patients in our study showed a strong correlation between functional deterioration and their neuropathic pain screening score. This study may be useful in understanding the characteristics of cervical radicular pain.

## 1. Introduction

Approximately 1 person in 1000 suffers from cervical radicular pain, which is most often caused by degenerative spondylosis or disc herniation [1]. Cervical radicular pain is pain perceived in the neck or upper limb, shooting or electric in quality resulting from nerve root dysfunction and is usually caused by a combination of the pressure exerted on the dorsal root ganglion and chemical irritation of the cervical nerve root, which leads to nerve root-specific dermatomal sensory or motor symptoms [1,2]. It is generally considered a mixed condition of nociceptive and neuropathic pain (NP). NP is defined as pain caused by a lesion or disease of the somatosensory nervous system [3]. The development of NP involves not only neuronal pathways but also Schwann cells, satellite cells in the dorsal root ganglia, components of the peripheral immune system, spinal microglia, and astrocytes [4]. It serves an important role in the pathogenesis of spinal pain [5].

The clinical distinction between the nociceptive and NP component in radicular pain is important, as it affects subsequent treatment decisions, and each has a different prognosis [6]. NP is often refractory to common analgesics and treatment, and patients with NP suffer from greater pain severity and medical costs, worse health, and relatively impaired quality of life compared to nociceptive pain [7,8]. However, estimation of the incidence and prevalence of NP in clinical practice has been difficult because of the lack of simple diagnostic criteria for large epidemiological surveys in the general population [9]. Accordingly, several symptoms-based screening questionnaires have been developed to identify patients with NP in order to alert clinicians for conducting further assessments, which have provided valuable new information on the prevalence of NP and are recommended for clinical use [10]. To date, the screening tools have increasingly been used and validated for the identification of an NP component in patients with low back pain and radicular leg symptoms [5,11,12,13,14,15,16]. However, few studies have quantified the relative proportions of nociceptive or NP components in cervicalgia compared with studies on low back pain. Moreover, neck pain may involve different patho-anatomical mechanisms [6,17]. This study aimed to determine the prevalence of NP components in patients with cervical radicular pain using screening tools and identify the relationship between NP components and clinical factors.

## 2. Materials and Methods

### 2.1. Study Population and Data Analysis

This study was approved by the Institutional Ethics Committee (HP IRB 2022-06-001) of the Inje University Haeundae Paik Hospital, Korea. We retrospectively reviewed the electronic medical records and radiologic data of 138 patients with cervical radicular pain acquired from a single pain clinic center between February 2020 and March 2022. Based on clinical evidence aided by magnetic resonance imaging or computed tomography, the patients were diagnosed with intervertebral herniated disc or spinal stenosis. Cervicalgia caused by a mechanical pathomechanism, referred to as pain originating in the spine or its supporting structures such as ligaments and muscles, was not considered. The inclusion criteria were patients aged ≥20 years, those who complained of cervical radicular pain, and a definitive diagnosis of herniated intervertebral disc or spinal stenosis. Data including sex, age, body mass index, history of diabetes or hypertension, history of neck surgery, initial pain intensity based on the numeric rating scale (NRS), dominant pain location, duration of symptoms, initial Douleur Neuropathique 4 questions (DN4), painDETECT questionnaire (PD-Q), and neck disability index (NDI) were assessed.

The initial pain intensity, divided into neck and upper limb pain, was measured using a 100 NRS with a score of 0 indicating no pain and a score of 100 indicating the worst possible pain. More severe pain locations were designated as those with dominant neck or arm pain. If there was no difference in pain intensity between the two locations, it was designated as a non-dominant pattern. Duration was defined as the time elapsed since the recent onset of an episode of pain, and the chronic state was defined as pain lasting for more than 12 weeks. The total score of the DN4 was calculated as the sum of 10 items, and the cutoff value for the screening of NP was a DN4 score of 4/10 [18]. The PD-Q comprises four blocks; by summing the scores in each block, a final score ranging from −1 to 38 can be obtained [19]. A score of ≥13 denotes the existence of a neuropathic component in pain. In the current study, the prevalence of an NP component in patients with cervical radicular pain was defined as when the results of both tests were positive.

The NDI used to assess health-related quality of life is a 10-item questionnaire scored using the percentage of the maximal pain and disability score [20]. Each item was scored between 0 and 5. In our study, the driving section was excluded, as several patients had missing information in this section. The maximum NDI score was 45. At our pain clinic, DN4 and PD-Q were investigated routinely in all patients, whereas NDI was assessed only in patients with neck pain. Thirty-five patients were excluded from the initial 138 cases because of incomplete medical records (Figure 1).

Finally, data from 103 patients (68 men and 35 women) were included for analysis in this study. The requirement for written informed consent was waived owing to the retrospective design of the study.

### 2.2. Statistical Analysis

Data are presented as frequencies with percentages for categorical variables and mean ± standard deviation for continuous variables. Differences in patient characteristics across the subgroups were compared using the chi-square test or Fisher’s exact test for categorical variables and the independent *t*-test or Mann–Whitney U test for continuous variables, as appropriate. To check for normal distribution, we used the Shapiro–Wilk test. The correlation between NDI and other variables was analyzed using Pearson’s correlation coefficient. Cohen’s kappa was used to assess the difference between the DN4 and PD-Q scores. Box plots are presented for data visualization. All statistical analyses were performed using SPSS (version 25.0; SPSS Inc., Chicago, IL, USA), and *p* values less than 0.05 were considered statistically significant.

## 3. Results

Clinical data were collected for all 103 patients included in this study (Table 1).

The dominant pain site was the arm in 48 (46.6%) patients. The mean pain intensities were 56.12 ± 27.45 for the neck, 68.54 ± 20.60 for the arm, and 71.07 ± 20.04 for the dominant pain site. When analyzing patients according to the combination of DN4 and PD-Q classifications, 29 patients (28.1%) were found to have NP components.

The mean NDI score in patients with cervical radicular pain was 19.94 ± 7.69. Between the patient groups, the NDI scores were significantly different (NP vs. non-NP group: 23.79 ± 6.35 vs. 18.43 ± 7.68, *p* < 0.001; Figure 2).

The DN4 or PD-Q and NDI scores were found to be significantly correlated (DN4: r = 0.221, *p* < 0.025; PD-Q: r = 0.368, *p* < 0.001; Table 2).

There were no significant differences in sex, age, BMI, comorbidities, pain duration, pain intensity, or dominant pain location between the two patient groups.

There was a high degree (82.5%) of agreement between the DN4 and PD-Q outcomes in identifying the NP component [*n* = 56 (no NP); *n* = 29 (NP); kappa 0.627, 95% confidence interval 0.437–0.781, *p* < 0.001]. For the 18 discordant cases, an NP component was demonstrated in only 13 patients by the DN4 but in five patients by the PD-Q (Table 3).

## 4. Discussion

Based on the combined positive results of DN4 and PD-Q, 28.1% of the patients in our study were found to predominantly have an NP component. Previously, Liu et al. [21] demonstrated that half of the 100 patients with chronic neck pain in their study had mixed pain conditions containing an NP component using the PD-Q and self-reported Leeds Assessment of Neuropathic Symptoms and Signs (s-LANSS) pain scale. Their study included participants with pain duration of 6 weeks or longer. Another study evaluating the effect of epidural injection in 67 individuals diagnosed with cervical disc herniation and having neck and radiating pain for at least 3 months found that 36.9% of them predominantly had NP components in accordance with s-LANSS [22]. The prevalence of an NP component was considerable in patients with neck or radiating pain, although direct comparisons with the prevalence of NP reported in other studies are limited because of the methodological heterogeneity. Although this limitation has also been reported in studies investigating the NP component in chronic low back pain [23], the scores of the screening questionnaires were found to be positively correlated with each other [16]. Although not many studies have focused on cervical radiculopathy, a few of them suggested comparable usefulness of the combined screening tools [12,21,24]. In our study, the combined positive results of DN4 and PD-Q were used to screen the NP component to achieve a high discriminant value. We found a substantial level of agreement between the two screening tools, DN4 and PD-Q, according to the criteria of Landis et al. [25]. This indicates congruency between their cutoff points. Bouhassira et al. have shown 83% sensitivity and 90% specificity of DN4 for the clinical diagnosis of NP [18]. In the study by Freynhagen et al., the PD-Q correctly classified 83% of patients in their diagnostic group, with a sensitivity of 85% and a specificity of 80% [19]. DN4 comprises both interview questions and physical examination, whereas PD-Q screens for typical signs and symptoms of NP, such as pain course patterns and the presence of radiating pain [23]. These are validated questionnaires to depict pain with a predominantly neuropathic origin and are offered as a screening tool in lumbar radiculopathy [5,11,12,13,14,15,16]. Further consensus on the classification and discrimination methods for cervical neuropathic pain is required.

The NDI is a valid questionnaire for measuring neck and arm pain related to disability in cervical radiculopathy [26]. Studies have shown patients in the NP group with higher disability scores at baseline compared with those with nociceptive pain [21,22]. In the correlation analysis of our study, the NDI score was positively correlated with the DN4 or PD-Q scores. This implies a higher NP screening tool score indicates increased debilitating effects on daily-life activities related to neck or arm pain in patients with cervical radicular pain. There was a stronger correlation between NDI and PD-Q (r = 0.368, *p* < 0.001) than that between NDI and DN4 (r = 0.221, *p* = 0.025). Thus, PD-Q is particularly useful for patients with spinal pain and radicular symptoms because it was developed to determine the degree of NP in patients with pain originating in the lumbar spine [7].

A previous study correlated the NP component with arm pain intensity but not neck pain [21,22]. This is possibly due to differences in the pain-generating mechanisms of each area. Moreover, the factors causing neck or upper-extremity pain are not clear to date. Direct mechanical compression of the dorsal root ganglion plays a dominant role in radiculopathy of far lateral disc herniation, with upper limb pain as the main symptom [27]. However, we could not identify a relationship between the pain intensity and the presence of NP in our study. Additionally, the differences in the dominant pain site were not significantly related to the NP component, although the dominant pain site for cervical radicular pain was the upper extremity. This difference may be due to the patient’s characteristics. Our study included 61.2% of patients in the acute pain stage with pain lasting for less than 12 weeks in the current episode period, unlike previous studies on patients with chronic pain. Cervical radicular pain intensity is most prominent in the acute phase and diminishes as the condition becomes more chronic [17]. Moreover, a longer disease duration ultimately progresses to neuropathic pain, which is a characteristic of a chronic state [28]. However, we did not confirm the relationship between pain duration and NP component, although a greater proportion of patients with chronic pain (32.5%) compared to acute pain (25.4%) appeared to have NP components. The patient’s remote pain was not a limitation due to the retrospective design of the current study.

This is one of the very few studies to evaluate the prevalence of NP in patients with cervical radicular pain. However, this study has a few limitations. First, it was designed as a retrospective chart review. Second, a relatively small number of patients have been included. To date, there has been no consensus regarding the standard diagnosis of NP for cervical radicular pain. Third, the DN4 and PD-Q assessments were not conducted blindly by independent examiners. Fourth, we did not consider patient medication information prior to testing.

## 5. Conclusions

In conclusion, this study showed that 28.1% of patients with cervical radicular pain had an NP component. These patients showed a strong correlation between functional deterioration and their NP screening score. Our findings may be useful for better understanding the pain characteristics and anticipating the clinical course of cervical radicular pain.

## Figures and Tables

**Figure 1 medicina-58-01191-f001:**
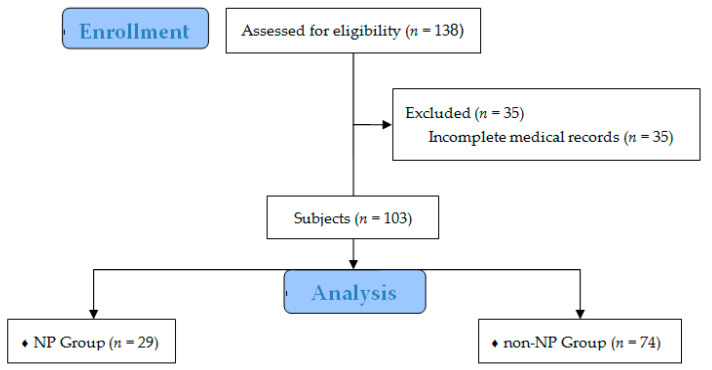
Flow diagram of the study. Abbreviation: NP, neuropathic pain.

**Figure 2 medicina-58-01191-f002:**
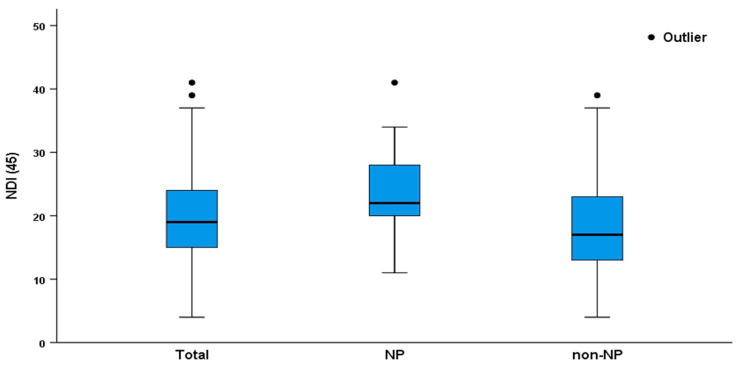
NDI in patients with cervical radicular pain. NP, neuropathic pain; NDI, neck disability index.

**Table 1 medicina-58-01191-t001:** Clinical findings of the patient groups.

Variables	Total (*n* = 103)	NP Component		*p* Value
		Yes (*n* = 29)	No (*n* = 74)	
Sex				
male	68 (66.0)	20 (69.0)	48 (64.9)	0.693 ^1^
female	35 (34.0)	9 (31.0)	26 (35.1)	
Age	56.55 ± 10.14	57.03 ± 9.36	56.36 ± 10.48	0.765 ^3^
Height	167.01 ± 7.28	168.48 ± 8.17	166.43 ± 6.88	0.200 ^3^
Weight	66.33 ± 10.01	67.66 ± 10.17	65.81 ± 9.97	0.308 ^4^
BMI	23.85 ± 3.07	24.26 ± 3.70	23.70 ± 2.80	0.521 ^4^
DM	15 (14.6)	2 (6.9)	13 (17.6)	0.223 ^2^
HTN	29 (28.2)	8 (27.6)	21 (28.4)	0.936 ^1^
OP history	9 (8.7)	4 (13.8)	5 (6.8)	0.265 ^2^
Duration				
acute	63 (61.2)	16 (55.2)	47 (63.5)	0.435 ^1^
chronic	40 (38.8)	13 (44.8)	27 (36.5)	
DN4	3.30 ± 2.04	5.66 ± 1.59	2.38 ± 1.33	<0.001 ^4^
PD-Q	10.17 ± 5.28	16.14 ± 4.29	7.84 ± 3.49	<0.001 ^4^
NDI (45)	19.94 ± 7.69	23.79 ± 6.35	18.43 ± 7.68	<0.001 ^4^
NRS neck	56.12 ± 27.45	61.38 ± 22.16	54.05 ± 29.14	0.252 ^4^
NRS arm	68.54 ± 20.60	70.00 ± 18.71	67.97 ± 21.39	0.651 ^4^
Dominant site				
arm	48 (46.6)	13 (44.8)	35 (47.3)	0.795 ^1^
neck	14 (13.6)	5 (17.2)	9 (12.2)	
non-dominant	41 (39.8)	11 (37.9)	30 (40.5)	
NRS dominant site	71.07 ± 20.04	73.10 ± 18.34	70.27 ± 20.74	0.474 ^4^

Values are expressed as mean ± standard deviation or absolute number (percentage). *p* values were derived by chi-square test ^1^, Fisher’s exact test ^2^, independent *t*-test ^3^, and Mann–Whitney U test ^4^. Abbreviations: NP, neuropathic pain; BMI, body mass index; DM, diabetes mellitus; HTN, hypertension; OP, operation; DN4, Douleure Neuropathique 4 questions; PD-Q, PainDETECT questionnaire; NDI, neck disability index; NRS, numeric rating scale.

**Table 2 medicina-58-01191-t002:** Correlation between DN4 or PD-Q score and NDI score in patients with cervical radicular pain.

	r	*p* Value
DN4	0.221	0.025
PD-Q	0.368	<0.001

Abbreviations: r, Pearson’s correlation coefficient; NDI, neck disability index; DN4, Douleure Neuropathique 4 questions; PD-Q, PainDETECT questionnaire.

**Table 3 medicina-58-01191-t003:** The frequency of NP component determined by the DN4 and PD-Q.

	DN4		
PD-Q	Positive	Negative	Total
Positive	29	5	34
Negative	13	56	69
Total	42	61	103

Abbreviations: NP, neuropathic pain; DN4, Douleure Neuropathique 4 questions; PD-Q, PainDETECT questionnaire.

## Data Availability

The data that support the findings of this study are available from the corresponding author upon reasonable request.

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
