# Peer review of "Neuropathic Pain Component in Patients with Cervical Radicular Pain: A Single-Center Retrospective Study"

_medicina, 2022, doi:10.3390/medicina58091191_

Round 1

Reviewer 1 Report

The main objective of the study is to define the prevalence of neuropathic pain in cervicobrachialgia, and this objective is achieved

The presence of neuropathic pain is significant in patients with scores above the specific cut-off levels of the DN4 and PD-Q scales. This does not seem like a very interesting finding since this fact has been used to define neuropathic pain. The DN4 and PD-Q scales were developed prior to defining diagnostic strength levels for neuropathic pain in 2008. The DN4 and PD-Q scales were not developed as diagnostic but as screening, so it could be interesting to assess whether there is a correlation between the results of these scales and the diagnostic confirmation of neuropathic pain.

Author Response

Manuscript ID:1839990

Title: Neuropathic Pain Component in Patients with Cervical Radicular Pain: a Single-center Retrospective Study

Review 1

The main objective of the study is to define the prevalence of neuropathic pain in cervicobrachialgia, and this objective is achieved

We thank the reviewers for your kind comments. Your comments and suggestions were helpful for revising our manuscript. We appreciate your careful review and look forward to receiving your feedback. Please note that the changes made do not influence the content, conclusions, or framework of the paper. We have not listed below all minor changes made; however, these are indicated in the revised manuscript.

The presence of neuropathic pain is significant in patients with scores above the specific cut-off levels of the DN4 and PD-Q scales. This does not seem like a very interesting finding since this fact has been used to define neuropathic pain. The DN4 and PD-Q scales were developed prior to defining diagnostic strength levels for neuropathic pain in 2008. The DN4 and PD-Q scales were not developed as diagnostic but as screening, so it could be interesting to assess whether there is a correlation between the results of these scales and the diagnostic confirmation of neuropathic pain.

Response: We agree with your comment. Neuropathic pain is a common debilitating problem in clinical practice. It is often refractory to common analgesics and treatment, and patients with neuropathic pain suffer from increased pain severity, greater costs, and relatively impaired quality of life. However, in most cases of chronic pain, it is difficult to establish the presence or absence of nerve dysfunction, regardless of symptoms (1). The estimation of the incidence and prevalence of neuropathic pain has been difficult because of the lack of simple diagnostic criteria. Several screening tools have been developed to identify patients who may have neuropathic pain to alert the clinician to undertake further assessment, although they cannot be used alone to confirm neuropathic pain. These screening tools discriminate patients with neuropathic pain from those with other types of chronic pain with up 80 % sensitivity and specificity. Especially, screening tools have been used for the identification of a NP component in patients with low back pain, although little research has been done on cervical radicular pain. Until a consensus is reached on a diagnostic approach to neuropathic pain, screening tools will serve to identify potential patients with neuropathic pain (2). We agree that it would be interesting to assess whether there is a correlation between the results of these scales and the diagnostic confirmation of neuropathic pain. According to our study design, we investigated the prevalence of a neuropathic pain component using established screening tools, and the combined results were used to screen the NP component to achieve a high discriminant value to complement this limitation. 

  1. Aggarwal VR, McBeth J, Zakrzewska JM, Lunt M, Macfarlane GJ. The epidemiology of chronic syndromes that are frequently unexplained: do they have common associated factors? Int J Epidemiol 2006;35:468–76.
  2. Bennett MI, Attal N, Backonja MM, Baron R, Bouhassira D, Freynhagen R, et al. Using screening tools to identify neuropathic pain. Pain. 2007;127:199-203.

Reviewer 2 Report

Thank you very much for allowing me to review this article. Please find my comments below.

ABSTRACT

-       Please try to avoid abbreviations as much as possible in the abstract.

-       Avoid the verb form "we”.

INTRODUCTION

-       The introduction is extremely brief. I suggest the authors develop this section further. Use other articles that discuss topics like yours. The length should be approximately one and a half pages.

METHODS

-       Forgive my ignorance, but... Why did the subjects have to be over 20 years old? In Europe the adult age is from 18 years of age, but I do not know if in your country this is the case. Thank you.

-       In the methodology section, I can hardly find any bibliographical references. It is necessary that all the sentences are justified with bibliography. For example, in the section "Study Population and Data Analysis" I cannot find a single reference.

I find the rest of the article adequate. Congratulations.

Author Response

Manuscript ID:1839990

Title: Neuropathic Pain Component in Patients with Cervical Radicular Pain: a Single-center Retrospective Study

Review 2

Thank you very much for allowing me to review this article. Please find my comments below.

We thank the reviewers for your kind comments. Your comments and suggestions were helpful for revising our manuscript. We appreciate your careful review and look forward to receiving your feedback. Please note that the changes made do not influence the content, conclusions, or framework of the paper. We have not listed below all minor changes made; however, these are indicated in the revised manuscript.

ABSTRACT

-       Please try to avoid abbreviations as much as possible in the abstract.

Response: We have revised the abbreviations accordingly. (Page 1)

-       Avoid the verb form "we”.

Response: We have revised the abstract accordingly. (Page 1)

INTRODUCTION

-       The introduction is extremely brief. I suggest the authors develop this section further. Use other articles that discuss topics like yours. The length should be approximately one and a half pages.

Response: We have revised the Introduction accordingly. (Pages 1, 2)

METHODS

-       Forgive my ignorance, but... Why did the subjects have to be over 20 years old? In Europe the adult age is from 18 years of age, but I do not know if in your country this is the case. Thank you.

Response: The adult standard in Korea has also recently changed, but it has not yet been fully applied in clinical practice. We also routinely applied the previous criteria to this study. Thank you very much for your point. We were not aware of this issue. The new standard will be applied in the next study.

-       In the methodology section, I can hardly find any bibliographical references. It is necessary that all the sentences are justified with bibliography. For example, in the section "Study Population and Data Analysis" I cannot find a single reference.

Response: We have revised the Methods accordingly. (Page 2–3, reference 18, 19, 20)

I find the rest of the article adequate. Congratulations.

Reviewer 3 Report

1.From the introduction literature: the literature review is not comprehensive and obsolete.

2.The font format should be strict and normal.

Author Response

Manuscript ID:1839990

Title: Neuropathic Pain Component in Patients with Cervical Radicular Pain: a Single-center Retrospective Study

Review 3

We thank the reviewers for your kind comments. Your comments and suggestions were helpful for revising our manuscript. We appreciate your careful review and look forward to receiving your feedback. Please note that the changes made do not influence the content, conclusions, or framework of the paper. We have not listed below all minor changes made; however, these are indicated in the revised manuscript.

  1. From the introduction literature: the literature review is not comprehensive and obsolete.

Response: We have revised the Introduction accordingly (Pages 1–2)

  1. The font format should be strict and normal.

Response: We have revised the format according to journal guideline.

Round 2

Reviewer 2 Report

Thank you for your answers. This article is suitable for your acceptance